# Vertical heterogeneity in predation pressure in a temperate forest canopy

Kathleen R. Aikens, Laura L. Timms and Christopher M. Buddle

Department of Natural Resource Sciences, McGill University, Ste-Anne-de-Bellevue, Quebec, Canada

## ABSTRACT

The forest canopy offers a vertical gradient across which variation in predation pressure implies variation in refuge quality for arthropods. Direct and indirect experimental approaches were combined to assess whether canopy strata differ in ability to offer refuge to various arthropod groups. Vertical heterogeneity in impact of avian predators was quantified using exclosure cages in the understory, lower, mid, and upper canopy of a north-temperate deciduous forest near Montreal, Quebec. Bait trials were completed in the same strata to investigate the effects of invertebrate predators. Exclusion of birds yielded higher arthropod densities across all strata, although treatment effects were small for some taxa. Observed gradients in predation pressure were similar for both birds and invertebrate predators; the highest predation pressure was observed in the understory and decreased with height. Our findings support a view of the forest canopy that is heterogeneous with respect to arthropod refuge from natural enemies.

## INTRODUCTION

Temperate forest canopies are heterogeneous environments, where variation in resources, structure, and abiotic conditions exists at even small scales (*Parker, 1995*). An important resource type that is often overlooked in ecological studies is the refuge – space within a habitat that allows organisms to escape from their natural enemies (*Berryman & Hawkins, 2006*). Vertical heterogeneity within forest canopies can offer refuge from predators, through the physical properties of the habitat as well as the foraging behaviour of both prey and natural enemies (*Jeffries & Lawton, 1984*). If refuge quality differs across a spatial gradient, predation then becomes an important determinant of local distribution.

Arthropods living in trees face strong predation pressure from vertebrate and invertebrate natural enemies (*Cornell & Hawkins, 1995*; *Mooney et al., 2010*). Predation can play an important role in shaping the niches of arthropods, particularly insect herbivores, by affecting the choice of feeding location, resource use, and ultimately fitness (*Jeffries & Lawton, 1984*; *Bernays & Graham, 1988*; *Stamp & Bowers, 1990*). These relationships, however, are poorly understood as a function of vertical stratification, even though it is well established that arthropods are structured along vertical gradients in forest (e.g., *Larrivée & Buddle, 2009*; *Pinzon, Spence & Langor, 2013*).

Corresponding author
Laura L. Timms,
laura.timms@mail.mcgill.ca

This research tested vertical heterogeneity in predation pressure in a north-temperate sugar maple (*Acer saccharum* Marsh.) forest to determine whether refuge from predators differs by canopy height. Optimal foraging theory predicts that predators should spend more time foraging in areas with higher prey density and reduced search time (*Emlen, 1966*; *MacArthur & Pianka, 1966*). These predictions are consistent with the observations of *Van Bael, Brawn & Robinson (2003)*, who found correspondingly higher arthropod abundance and predation pressure in the tropical forest canopy vs. understory. Previous work in our system suggests arthropod abundance decreases with distance from the forest floor (*Aikens & Buddle, 2012*), which would mean higher pay-off for predators foraging in the understory and lower canopy layers and thus higher predation pressure for arthropods in these strata. Furthermore, increased density and structural complexity of foliage in the upper canopy may provide more camouflage for arthropod prey and increase predator search time. We therefore predict that the upper canopy crown represents a refuge for arthropods and that the relative impact of predators will be reduced with increasing canopy height.

## MATERIALS AND METHODS

### Study site

Experiments were completed at the Morgan Arboretum, a 245-hectare forest reserve near Montréal (Ste-Anne-de-Bellevue, Québec, Canada, (45° 26′ N, 73° 57′ W)). The Arboretum contains tracts of natural woodland and collections of exotic trees, although most of the forested area is typical temperate beech (*Fagus grandifolia* (Ehrh.)) and sugar maple forest. Natural stands of sugar maple were selected for this study, including mature trees with heights of approximately 20–25 m. Trees were selected on the basis of accessibility of the canopy using a mobile aerial lift platform.

To examine differences in predation on arthropods along a vertical gradient, we carried out manipulations in four vertical strata: understory, lower canopy, middle canopy, and upper canopy. We defined strata in relative terms rather than in absolute height because of variance in both absolute tree height and depth of canopy foliage. The lower canopy was defined as the first several layers of branches encountered (~10–12 m); the middle canopy as the layers of branches at the midpoint of total tree foliage (~15–17 m); and the upper canopy as the several layers of branches at the very top of the foliage (~20–25 m). The understory was defined as the first two meters above the forest floor; which, because mature sugar maples do not have foliage at these heights, meant that the understory stratum was composed of sugar maple saplings. We acknowledge that this has the potential to add ontogeny as a confounding factor to our study. However, a recent meta-analysis found no overall preference for insect herbivores between saplings and mature trees, despite significant ontogenetic changes in leaf chemistry (*Barton & Koricheva, 2010*).

### Bird predation

We used wire exclosure cages to assess differences in vertebrate predation across strata (Fig. 1). Cages had a mesh diameter of 2.5 × 3.5 cm, which excludes the majority of

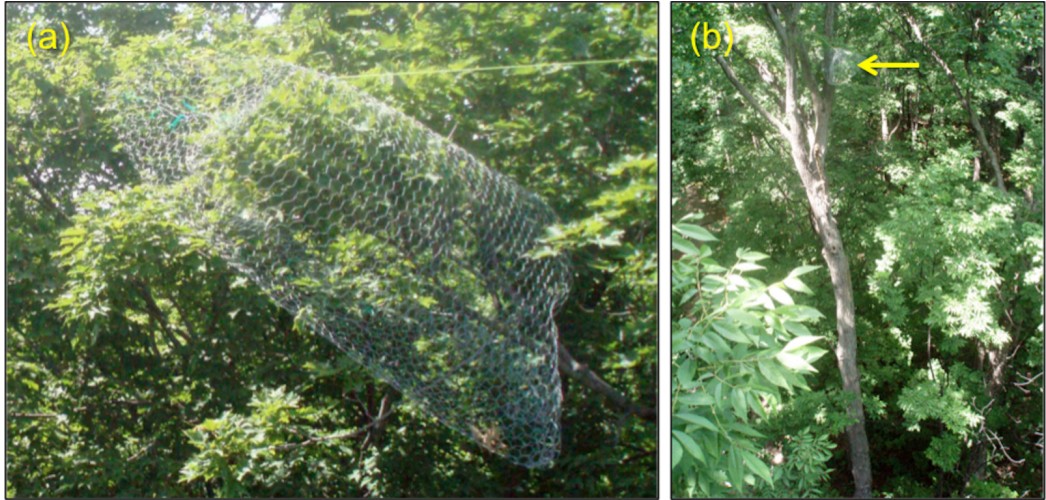

**Figure 1 Exclusion cage design and placement.** (A) Example of a wire mesh cage used to exclude vertebrate predators, (B) location of cage in the middle canopy of a tree.

vertebrate predators (mainly birds) while still allowing access to most insects. Similar cages have been used elsewhere to assess the effects of predation on arthropod abundance (e.g., *Marquis & Whelan, 1994*; *Boege & Marquis, 2006*). Each cage measured approximately one meter in length, with a circumference of $1.7 \pm 0.11$ m (mean $\pm$ SE). Cage design and placement did not compress leaves or deform branches in any way; cages were supported with ropes from above to relieve weight and pressure on the branches and leaves within the enclosure (Fig. 1A). We selected 20 mature trees and attached a single cage on one branch in each stratum, for a total of 20 cages per stratum. Sugar maple saplings nearest each tree were selected for understory exclosures. Each cage branch was paired with a control branch of similar size.

We completed arthropod censuses at intervals of approximately two weeks, for a total of six sampling periods from June to September 2007. We collected arthropods by shaking each branch onto a stretched canvas collecting frame, identifying all arthropods to the most precise taxonomic level possible and then returning them to the branch. Flies (Diptera), bees and wasps (Hymenoptera, excluding ants), and some groups of true bugs (e.g., Cercopidae) were excluded from collections, as they are too mobile to be accurately censused. All cage and control branches were removed in September, and the dry mass of leaves on each branch was measured. Comparisons between cage and control branches are thus expressed as arthropod density per unit leaf biomass.

We tested the effects of vertical stratum and vertebrate exclusion on arthropod density using linear mixed effects models, with stratum and treatment as fixed effects and tree identity (including one understory sapling and the three canopy levels of one mature tree) as a random effect. Modeling tree identity as a random effect within a mixed model allowed us to account for the fact that the data include multiple observations from each of the twenty tree pairs (*Lindstrom & Bates, 1990*; *Zuur et al., 2009*). Response variables included the density of (i) all arthropods, (ii) spiders (Araneae), (iii) beetles (Coleoptera),

(iv) true bugs (Hemiptera), and (v) caterpillars (Lepidoptera), pooled over the six collection periods. Arthropod densities were square-root transformed before analysis to increase normality. One understory cage and one mid-canopy cage from the same tree were lost late in the season, and were excluded from analyses. Model fitting and checking procedures were carried out as recommended in *Zuur et al. (2009)*, including: fitting the full model; finding the optimal random structure; and, finding the optimal fixed structure. The fit of these models to the data was assessed using diagnostic graphical methods, including plots of fitted values versus standardized residuals as well as normal QQ plots for both the fixed and random residual error. Contrasts were defined for each model using backwards difference coding, such that the means for each stratum were compared to the stratum before it – in other words, the lower canopy mean was compared to the understory mean, the middle canopy mean was compared to the lower canopy mean, and the upper canopy mean was compared to the middle canopy mean.

We tested the hypothesis that predation pressure would differ between vertical strata using the interaction between exclusion treatment and stratum. However, a non-significant interaction does not exclude the possibility that the magnitude of treatment effects differs between strata (e.g., *Boege & Marquis, 2006*). Thus, we calculated effect sizes (Cohen's d) and their 95% confidence intervals for the vertebrate exclusion treatment in each stratum and for each response group. Means for each stratum-treatment combination were calculated based on the season total density for each branch of that type ($n = 20$). An effect size was considered to be non-significant if its confidence intervals overlapped the value zero. All statistical analyses were carried out in R (*R Core Team, 2012*), with the use of the packages nlme (*Pinheiro et al., 2012*) and lme4 (*Bates, Maechler & Bolker, 2012*).

## Bird census

Birds were censused within the areas selected for exclosure experiments during mid-May to early June 2007. Five sites were chosen as stationary points, and all individuals within a 100 m radius were identified to species by sight or vocalization. We delimited the sites in this way to fully census the experimental area while reducing the potential of recording individuals residing in nearby fields. Censuses occurred over ten days, beginning at dawn and ending approximately 2.5–3 h later. A total of 20.9 survey hours were completed. Efforts were made to ensure that no individual was recorded twice during the same day; however, we cannot guarantee that the same individuals were not counted on multiple days. Therefore, data represent relative frequencies of species and not abundance data.

## Invertebrate predation

We used bait trials to assess differences in invertebrate predation across vertical strata (see *Olson, 1992*; *Novotny et al., 1999*). These trials were completed on 21 sugar maple trees during June and July 2007. In each trial immobilized, but live, mealworm larvae (*Tenebrio molitor* L.) were pinned to the bark of the tree as bait. Five bait mealworms were fastened to the bark in each stratum by puncturing their body with an insect pin and securing the pin into bark. Though these beetle larvae are not naturally found in the forest, they represent a

generally palatable prey for many arthropod and bird species (e.g., *Uetz, Bischoff & Raver, 1992*; *Vivan, Torres & Veiga, 2003*; *Grieco, 2003*). We chose to use mealworms for these trials as it allowed us to estimate a relative value of predation pressure, comparable across strata and free from idiosyncrasies of prey defense chemicals or behavior. We acknowledge that this does not correspond to predation pressure experienced by any one naturally occurring prey species; however, mealworms are acceptable prey for many generalist predators (e.g., *Lymberry & Bailey, 1980*).

All trials were completed between 0900 and 1700 h, during which period baits were checked a total of four times over 150 min. The first observation was recorded after 60 min, and each successive observation was taken 30 min after the previous one. During each observation period, predator presence and type (e.g., ant, jumping spider, fly, etc.) were recorded. Missing mealworms were also recorded as depredated, as it was observed that large ants (*Camponotus* sp.) were capable of removing mealworms from the pin within minutes. After initial observation, baits were left overnight and examined once more the following morning.

We tested the effects of vertical stratum on the proportion of predated mealworms using binomial generalized linear mixed models, with stratum as a fixed effect and tree identity as a random effect. Model fitting and checking procedures were carried out as outlined in *Bolker et al. (2009)*, including: fitting the full model; checking for overdispersion; finding the optimal random structure; and, finding the optimal fixed structure. A priori contrasts between strata were defined as described above. We repeated this analysis both for the proportion of predated mealworms observed during the initial 150 min as well as for the proportion removed after the overnight period.

## RESULTS

### Vertebrate predation

More than 2,600 individual arthropods identified from 49 families were surveyed from June to September 2007. Spiders were the most abundant group (57%), followed by beetles (16%), true bugs (10%), and caterpillars (9%). Other arthropod groups, including ants (Hymenoptera: Formicidae), earwigs (Dermaptera), harvestmen (Opiliones), and booklice (Psocoptera) were observed in small numbers (Table 1).

Selection procedures for the mixed effects models found that the best model for all response groups included a weighted variance structure due to differences in residual variances between strata. Exclusion treatment and vertical stratum were significantly related to arthropod density for all response groups, but the interaction between the two was not significant for any response group (Table 2). Mean arthropod density was higher on caged branches than on controls, and in general decreased with increasing height (Fig. 2). Density was highest in the understory for all groups except caterpillars, whose density was highest in the lower canopy; the results of comparisons between strata are presented in Table 2.

Based on effect sizes, the magnitude of the vertebrate exclusion treatment effect on density was largest in the understory in the analyses for all arthropods, spiders, and

**Table 1 Total individuals collected, by order and family.** Arthropods were collected during branch beating surveys in 20 sugar maple (*Acer saccharum* Marsh.) trees in four vertical strata (UN, understory; LC, lower canopy; MC, middle canopy; UC, upper canopy) in the Morgan Arboretum during June–September 2007; rank provides information on the total abundance of families.

| Order | Family | UN | LC | MC | UC | TOTAL | Rank |
|---|---|---|---|---|---|---|---|
| **Araneae** | Agelenidae | 6 | 0 | 0 | 0 | 6 | 26 |
| | Araneidae | 63 | 34 | 26 | 16 | 139 | 5 |
| | Clubionidae | 37 | 36 | 41 | 29 | 143 | 4 |
| | Dictynidae | 15 | 11 | 8 | 4 | 38 | 16 |
| | Linyphiidae | 24 | 9 | 5 | 7 | 45 | 15 |
| | Philodromidae | 12 | 48 | 25 | 19 | 104 | 7 |
| | Salticidae | 216 | 155 | 153 | 263 | 787 | 1 |
| | Tetragnathidae | 4 | 0 | 0 | 0 | 4 | 27 |
| | Theridiidae | 59 | 8 | 4 | 3 | 74 | 9 |
| | Thomisidae | 22 | 1 | 2 | 4 | 29 | 17 |
| | Unidentified | 48 | 39 | 33 | 12 | 132 | . |
| **Coleoptera** | Bostrichidae | 2 | 2 | 4 | 0 | 8 | 25 |
| | Buprestidae | 1 | 1 | 2 | 4 | 8 | 25 |
| | Cantharidae | 1 | 9 | 4 | 0 | 14 | 22 |
| | Carabidae | 1 | 0 | 0 | 0 | 1 | 30 |
| | Cerambycidae | 2 | 0 | 1 | 0 | 3 | 28 |
| | Chrysomelidae | 8 | 1 | 1 | 1 | 11 | 24 |
| | Cleridae | 0 | 1 | 1 | 0 | 2 | 29 |
| | Coccinellidae | 27 | 10 | 5 | 17 | 59 | 10 |
| | Curculionidae | 20 | 11 | 12 | 6 | 49 | 13 |
| | Elateridae | 2 | 0 | 0 | 0 | 2 | 29 |
| | Lampyridae | 18 | 6 | 3 | 0 | 27 | 18 |
| | Latridiidae | 2 | 7 | 2 | 1 | 12 | 23 |
| | Oedomeridae | 2 | 0 | 1 | 0 | 3 | 28 |
| | Pyralidae | 0 | 1 | 2 | 1 | 4 | 27 |
| | Scirtidae | 10 | 1 | 0 | 0 | 11 | 24 |
| | Staphylinidae | 1 | 9 | 7 | 1 | 18 | 21 |
| | Tenebrionidae | 29 | 56 | 45 | 23 | 153 | 3 |
| | Unidentified | 11 | 8 | 18 | 7 | 44 | . |
| **Dermaptera** | Forficulidae | 4 | 21 | 19 | 9 | 53 | 11 |
| **Hemiptera** | Aphididae | 4 | 13 | 4 | 3 | 24 | 19 |
| | Cercopidae | 0 | 0 | 0 | 1 | 1 | 30 |
| | Lygaeidae | 0 | 1 | 0 | 0 | 1 | 30 |
| | Miridae | 4 | 7 | 6 | 3 | 20 | 20 |
| | Nabidae | 1 | 2 | 1 | 0 | 4 | 27 |
| | Pentatomidae | 8 | 10 | 16 | 14 | 48 | 14 |
| | Reduviidae | 40 | 35 | 37 | 47 | 159 | 2 |
| | Thyreocoridae | 0 | 0 | 1 | 0 | 1 | 30 |
| | Unidentified | 1 | 3 | 6 | 4 | 14 | . |
| **Hymenoptera** | Formicidae | 57 | 15 | 3 | 2 | 77 | 8 |

Table 1 (*continued*)

| Order | Family | UN | LC | MC | UC | TOTAL | Rank |
|-------|--------|----|----|----|----|-------|------|
| **Lepidoptera** | Geometridae | 21 | 42 | 38 | 21 | 122 | 6 |
| | Lasiocampidae | 0 | 2 | 1 | 0 | 3 | 28 |
| | Limacodidae | 0 | 2 | 0 | 1 | 3 | 28 |
| | Lymantriidae | 0 | 1 | 2 | 1 | 4 | 27 |
| | Noctuidae | 3 | 8 | 1 | 2 | 14 | 22 |
| | Papillionidae | 0 | 0 | 1 | 0 | 1 | 30 |
| | Pyralidae | 1 | 5 | 2 | 0 | 8 | 25 |
| | Tortricidae | 2 | 17 | 18 | 14 | 51 | 12 |
| | Unidentified | 1 | 10 | 13 | 3 | 27 | · |
| **Odonata** | · | 1 | 0 | 0 | 0 | 1 | 30 |
| **Opiliones** | · | 31 | 11 | 6 | 3 | 51 | · |
| **Psocoptera** | · | 0 | 0 | 0 | 1 | 1 | 30 |
| **Trichoptera** | · | 1 | 0 | 0 | 0 | 1 | 30 |
| | **TOTAL** | **824** | **669** | **580** | **547** | **2619** | |

beetles; however, for beetles the effect was not significantly different than zero as the 95% confidence intervals just overlap the zero line (Fig. 3). None of the effect sizes were significantly different than zero for true bugs or caterpillars (Fig. 3).

Spring bird censuses yielded a total of 31 species observed or heard vocalizing in and around experimental trees (Table 3). The most frequently recorded species were American Crow (*Corvus brachyrhynchos*), American Goldfinch (*Carduelis tristis*), Red-eyed Vireo (*Vireo olivaceus*), Ovenbird (*Seiurus aurocapillus*), Black-throated Green Warbler (*Setophaga virens*), and Black-capped Chickadee (*Poecile atricapillus*). The latter four species are primarily insectivorous in the spring and early summer (*Holmes, Bonney & Pacala, 1979*; *De Graaf, Tilghman & Anderson, 1985*).

## Invertebrate predation

Of 840 mealworm baits, 134 were removed or observed with feeding invertebrate predators. A total of 194 predators were observed, the majority of which were ants (Formicidae) (157, or 81%). Also recorded were: 17 muscoid flies (Diptera: Muscidae, Tachinidae, Calliphoridae, Sarcophagidae); 13 harvestman (Opiliones: Phalangidae, Sclerosomatidae); 3 jumping spiders (Araneae: Salticidae); and 3 predatory bugs (Hemiptera: Reduviidae). Predatory arthropod species that could be reliably identified in the field are listed in Table 4. The highest number of predators was observed in the understory; however, after excluding a large number of ants (60) observed on a single occasion in the understory, the highest number of observed predators was seen in the lower canopy.

The highest proportion of observed daytime predation occurred in the understory, followed by the lower canopy, upper canopy, and middle canopy (Fig. 4A). Predation in the middle canopy was significantly lower than both the lower canopy and the upper canopy, however, predation in the lower canopy was not significantly lower than the understory (Table 5). The highest proportion of overnight removal of baits also occurred in the understory, followed by lower canopy, middle canopy, and upper canopy (Fig. 4B).

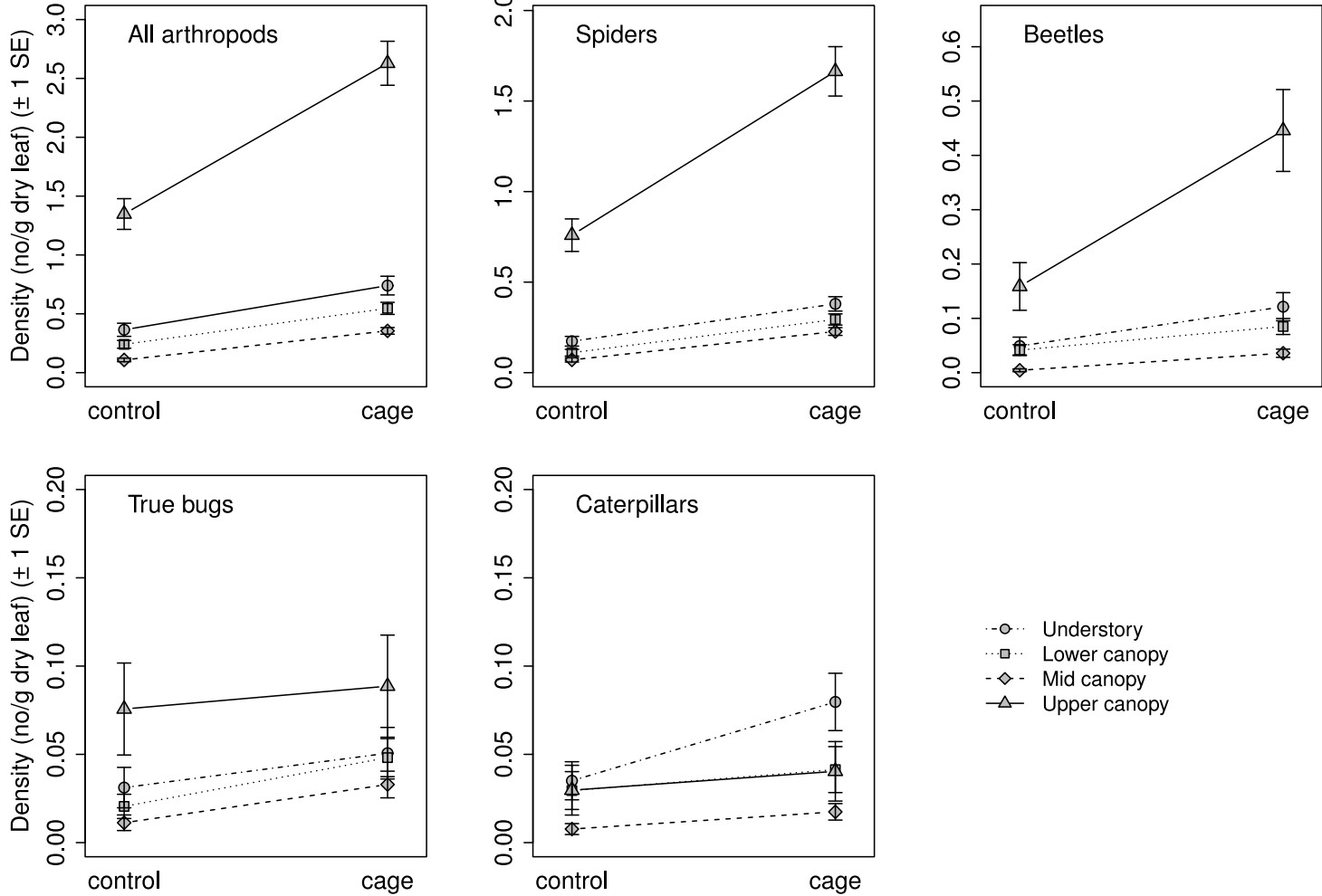

**Figure 2 Arthropod density by treatment and stratum.** Mean density (± SE) on control and caged branches from 20 sugar maple (*Acer saccharum* Marsh.) trees across four vertical strata: understory (UN); lower canopy (LC); middle canopy (MC); and upper canopy (UC). Means and standard errors are back-transformed predicted values from mixed effects models with tree identity as a random effect.

The lower canopy had significantly lower proportions of overnight bait removal than the understory, however neither of the other stepwise comparisons showed significant differences (Table 5).

## DISCUSSION

Excluding birds and other vertebrates in our cage treatment resulted in increased density of all study groups, indicating that vertebrate predation has a significant negative impact on arthropod density. In addition, our results demonstrate vertical heterogeneity in both density of arthropods and some aspects of predation pressure on arthropods in mature sugar maple trees. In particular, the understory contained the greatest density of arthropods, showed the greatest predation rates by invertebrate natural enemies, and presented the highest vertebrate predation pressure on spiders and beetles. However, we found no differences in predation pressure between strata for the other arthropod groups in our study.

**Table 2  Results of mixed effects models testing effects of predator exclusion and vertical stratum.**
Models tested the fixed effects of predator exclusion (cage vs. control) and vertical stratum (UN, understory; LC, lower canopy; MC, middle canopy; UC, upper canopy) on the density of various groups of arthropods surveyed from June–September 2007 on 20 sugar maple trees (*Acer saccharum* Marsh.), while considering the random effects of tree identification (not shown); comparisons between vertical strata are presented as stepwise contrasts. Bold text indicates significant *p*-values.

| Response group | Factor | DF | F | *p* | Comparisons |
|---|---|---|---|---|---|
| All arthropods | Treatment | 1, 131 | 174.98 | **<0.0001** | Cage > Control[****] |
| | Stratum | 3, 131 | 172.95 | **<0.0001** | LC < UN[****] |
| | Trt[*] Strat | 3, 131 | 1.98 | 0.1196 | MC < LC[**] |
| | | | | | UC < MC[****] |
| Spiders | Treatment | 1, 131 | 123.90 | **<0.0001** | Cage > Control[****] |
| | Stratum | 3, 131 | 106.55 | **<0.0001** | LC < UN[****] |
| | Trt[*] Strat | 3, 131 | 2.49 | 0.0635 | MC < LC[**] |
| | | | | | UC < MC[**] |
| Beetles | Treatment | 1, 131 | 47.55 | **<0.0001** | Cage > Control[****] |
| | Stratum | 3, 131 | 43.16 | **<0.0001** | LC < UN[****] |
| | Trt[*] Strat | 3, 131 | 1.55 | 0.2044 | MC < LC |
| | | | | | UC < MC[****] |
| True bugs | Treatment | 1, 131 | 11.76 | **0.0008** | Cage > Control[***] |
| | Stratum | 3, 131 | 5.44 | **0.0015** | LC < UN[*] |
| | Trt[*] Strat | 3, 131 | 0.24 | 0.8681 | MC < LC |
| | | | | | UC < MC |
| Caterpillars | Treatment | 1, 131 | 7.98 | **0.0055** | Cage > Control[**] |
| | Stratum | 3, 131 | 11.04 | **<0.0001** | LC > UN |
| | Trt[*] Strat | 3, 131 | 0.53 | 0.6638 | MC < LC |
| | | | | | UC < MC[**] |

**Notes.**

Comparisons between strata were conducted using contrasts with backwards difference coding, significance of comparisons ($t_{134}$) designated by

[****] $p < 0.0001$.

[***] $p < 0.001$.

[**] $p < 0.01$.

[*] $p < 0.05$.

The densities of most arthropod taxa in our study were highest in the understory and decreased with increasing canopy height. These results are consistent with the majority of descriptions of the vertical stratification of arthropods in temperate forests (e.g., *Preisser, Smith & Lowman, 1998*; *Larrivée & Buddle, 2009*; *Ulyshen, 2011*), although it has been noted that certain species and trophic groups that we did not survey (e.g., some parasitoids and predatory wasps) are more abundant in the upper canopy (e.g., *Vance et al., 2007*; *Sobek et al., 2009*). The general pattern of higher abundance and richness of temperate forest arthropods near to the forest floor has been explained by a number of factors, including the greater stability of the microclimate nearer to the ground (*Parker, 1995*)

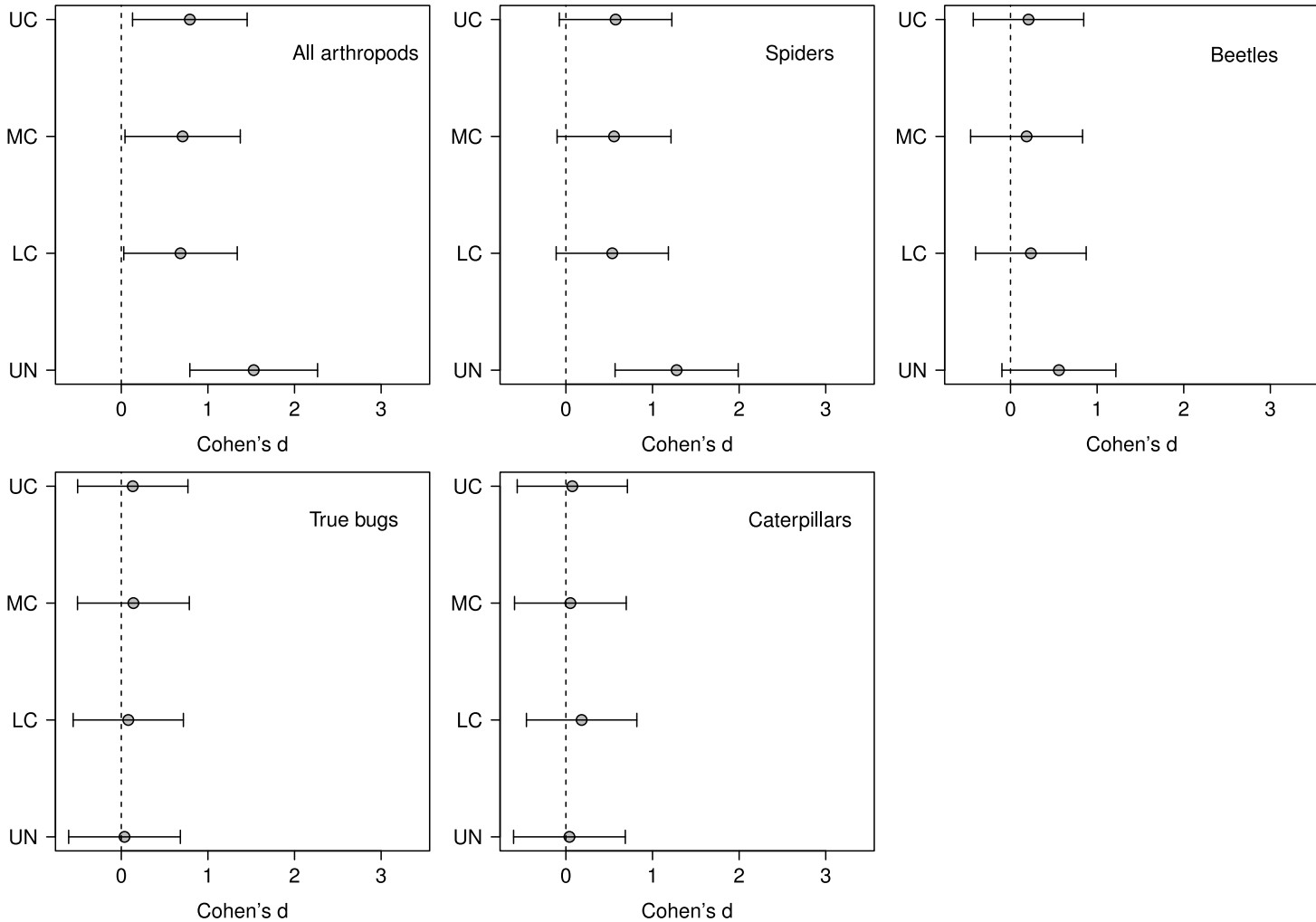

**Figure 3  Effect sizes by stratum for predator exclusion treatment.** Points represent the magnitude of the effects of a predator exclusion treatment on arthropod density surveyed from June–September 2007 on branches from 20 sugar maple (*Acer saccharum* Marsh.) trees in four vertical strata: understory (UN); lower canopy (LC); middle canopy (MC); and upper canopy (UC). Effect sizes and 95% CIs were calculated using predicted values from mixed effects models with tree identity as a random effect.

and dispersal limitation after emergence (*Brown et al., 1997*). Refuge from natural enemies and the distribution and quality of food resources may also explain the pattern.

Seasonality in the temperate forest means that most arthropods in these systems likely overwinter at or below ground level, and must recolonize the newly grown canopy habitat every spring. When other factors do not exert strong selection pressure to move further up the tree, this factor alone can provide a reasonable explanation for the greater abundance and diversity of temperate forest arthropods at lower heights. For example, after ruling out a number of other possible factors, including foliage quality, microclimate, leaf phenology, and natural enemies, *Brown et al. (1997)* concluded that dispersal limitation following spring emergence is the best explanation for the high densities of a leaf mining moth in the lower canopy. *Ulyshen (2011)* notes that many predators of forest arthropods are generally more abundant in the upper canopy, including birds, parasitoids, and predatory wasps.

**Table 3 Bird species recorded during census periods.** Birds were recorded by sight or by vocalizations from May 28–June 5, 2007 in sugar maple (*Acer saccharum* Marsh.) stands of the Morgan Arboretum, within 100 m of plots used for predator exclusion trials. Percent relative abundance was calculated as the number of records for a given species divided by the total number of records.

| Common name | Scientific name | Records | Rel. abd. (%) |
|---|---|---|---|
| Yellow-bellied Sapsucker | *Sphyrapicus varius* | 21 | 5.7 |
| Downy Woodpecker | *Picoides pubescens* | 2 | 0.5 |
| Northern Flicker | *Colaptes auratus* | 5 | 1.4 |
| Pileated Woodpecker | *Dryocopus pileatus* | 5 | 1.3 |
| Eastern Wood-Peewee | *Contopus virens* | 8 | 2.2 |
| Great-crested Flycatcher | *Myiarchus crinitus* | 14 | 3.8 |
| Empidonax Flycatcher | *Empidonax* sp. | 1 | 0.3 |
| Philadelphia Vireo | *Vireo philadelphicus* | 18 | 4.9 |
| Warbling Vireo | *Vireo gilvus* | 1 | 0.3 |
| Red-eyed Vireo | *Vireo olivaceus* | 39 | 10.5 |
| Bluejay | *Cyanocitta cristata* | 10 | 2.7 |
| American Crow | *Corvus brachyrhynchos* | 44 | 11.9 |
| Black-capped Chickadee | *Poecile atricapillus* | 26 | 7.0 |
| White-breasted Nuthatch | *Sitta carolinensis* | 1 | 0.3 |
| Veery | *Catharus fuscescens* | 7 | 1.9 |
| American Robin | *Turdus migratorius* | 3 | 0.8 |
| Northern Parula | *Setophaga americana* | 15 | 4.1 |
| Yellow Warbler | *Setophaga petechia* | 1 | 0.3 |
| Chestnut-sided Warbler | *Setophaga pensylvanica* | 12 | 3.2 |
| Magnolia Warbler | *Setophaga magnolia* | 1 | 0.3 |
| Cape May Warbler | *Setophaga tigrina* | 1 | 0.3 |
| Black-throated Green Warbler | *Setophaga virens* | 27 | 7.3 |
| Warbler sp. | *Setophaga* sp. | 2 | 0.5 |
| Black-and-white Warbler | *Mniotilta varia* | 2 | 0.5 |
| Ovenbird | *Seiurus aurocapilla* | 39 | 10.5 |
| Scarlet Tanager | *Piranga olivacea* | 1 | 0.3 |
| Rose-breasted Grosbeak | *Pheuticus ludovicianus* | 4 | 1.1 |
| Northern Cardinal | *Cardinalis cardinalis* | 3 | 0.8 |
| Chipping Sparrow | *Spizella passerina* | 1 | 0.3 |
| Red-winged Blackbird | *Agelaius phoeniceus* | 9 | 2.4 |
| American Goldfinch | *Carduelis tristis* | 47 | 12.7 |

These two factors might combine to create strong selection for arthropods to forage in the lower canopy or understory, unless there is a large trade-off in resource quality.

Foliage within a tree can vary in suitability for herbivorous insects, such that leaf age, size, and position can affect the nutritional quality and quantity of secondary chemicals. Herbivorous insects are commonly thought to optimize their foraging in order to maximize nutrition and minimize attack by natural enemies (e.g., *Carroll & Quiring, 1994*). However, there are many exceptions to this rule; for example, spruce budworm

**Table 4 Identifiable arthropod predators observed during bait trials.** Predators were observed feeding on mealworm larva baits in the Morgan Arboretum during trials conducted in June and July 2007, plus three predatory ant species collected from foliage on the same trees but not directly observed feeding on baits.

| Order | Family | Species |
|---|---|---|
| Araneae | Salticidae | *Eris militaris* |
| Opiliones | Phalangidae | *Odiellus pictus* |
| | Sclerosomatidae | *Leiobunum aldrichi* |
| Hemiptera | Reduviidae | *Zelus luridus* |
| Hymenoptera | Formicidae | *Camponotus pennsylvanicus* |
| | | *Aphaenogaster* spp.[*] |
| | | *Lasius alienus*[*] |
| | | *Leptothorax longispinosus*[*] |

**Notes.**

[*] These ant species were not directly observed predating mealworm larvae, but were collected from foliage on the same trees and are known to feed on arthropods.

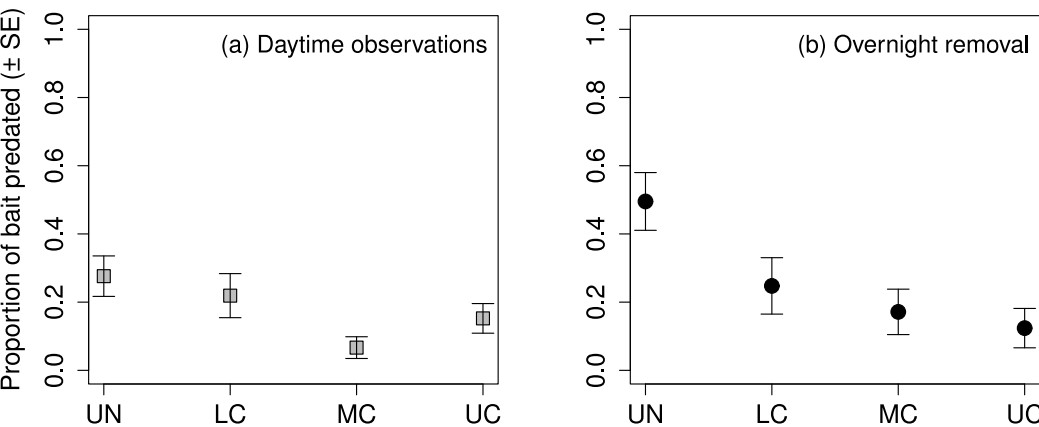

**Figure 4 Proportion of mealworm bait predated by stratum.** Proportion of mealworm larvae bait predated (±SE) on sugar maple (*Acer saccharum* Marsh.) branches (A) during daytime observations, and (B) removed after an overnight period, in four vertical strata: understory (UN); lower canopy (LC); middle canopy (MC); and upper canopy (UC).

larvae prefer to feed on the nutritious foliage in the upper canopy of host trees, which is also where they experience the highest parasitism rates and pathogen infection (*Dodge, 1961*; *Régnière, Lysyk & Auger, 1989*). Caterpillars in our study were the only group whose density was highest above the understory, although the overall low numbers of caterpillars observed make comparisons difficult. For the other herbivorous groups in our study the high densities observed in the understory may be a result of the quality or diversity of food sources as well as the dilution effect, where individual predation risk decreases as the number of potential prey goes up (e.g., *Finkbeiner, Briscoe & Reed, 2012*).

The majority of the individuals observed in our study are generalist predators of arthropods (i.e., spiders, predatory bugs), and a large part of their within-tree distribution

**Table 5 Results of binomial glm testing the effects of vertical stratum on predation.** Models tested the fixed effects of vertical stratum (UN, understory; LC, lower canopy; MC, middle canopy; UC, upper canopy) on the proportion of mealworm bait predated during two trial periods, while considering the random effects of tree identification (not shown); comparisons between vertical strata are presented as stepwise contrasts. Bold text indicates significant *p*-values.

| Predation period | Comparison | Est. | SE | Z | *p* |
|---|---|---|---|---|---|
| Daytime | LC < UN | −0.37 | 0.35 | −1.07 | 0.2849 |
| | MC < LC | −1.59 | 0.49 | −3.20 | **0.0014** |
| | UC > MC | −1.06 | 0.51 | 2.06 | **0.0397** |
| Overnight | LC < UN | −1.49 | 0.35 | −4.26 | **<0.0001** |
| | MC < LC | −0.60 | 0.39 | −1.55 | 0.1220 |
| | UC < MC | −0.54 | 0.45 | −1.20 | 0.2310 |

may therefore be influenced by where the bulk of their prey is located. Predators can be flexible in their habitat use, and some species will track the distribution of their prey through space (*Murakami, 2002*). We also note that spider density increased substantially within cages, and it is possible that increased spider predation may have reduced the effect of the cage treatment for other groups. When intermediate predators are released from predation pressure, populations may increase to the point where they control herbivores more effectively than higher order predators (*Tscharntke, 1997*), and the removal of vertebrate predators may therefore not always result in decreases of invertebrate herbivore density. However, a recent meta-analysis of predator exclusion studies found that exclusion of vertebrate predators reduced the density of invertebrate predators and herbivores by the same magnitude, indicating that the release of vertebrate predation pressure on invertebrate predators did not result in increased predation by invertebrate predators on herbivores (*Mooney et al., 2010*). This may be due to intraguild predation, where predators prey on each other; for example, it is estimated that about 20% of the diet of hunting spiders is made of other spiders (*Hodge, 1999*).

Although we found that vertebrate predator exclusion was significantly related to increases in density for all response groups, it is worth noting that the effect size was only significantly different from zero for spider densities in the understory, and not for any other individual response group or stratum (Fig. 3). Spiders were also the only response group in which the interaction between treatment and stratum was close to significant (Table 2). In their meta-analysis *Mooney et al. (2010)* found that caterpillars and spiders exhibited the two strongest responses in density to predator exclusion (based on effect sizes), followed by Hemiptera, Coleoptera, and Hymenoptera. The consistent and strong effect of bird exclusion on caterpillars in the meta-analysis reflects the fact that caterpillars are usually a favoured prey item of birds (*Robinson & Holmes, 1982*; *Marshall et al., 2002*). However, during years and periods of the season when caterpillar abundance is low most birds will switch to feeding on other arthropods, including beetles and true bugs, rather than change foraging locations (*Sample, Cooper & Whitmore, 1993*; *Murakami, 2002*). In our study, the peak caterpillar density occurred in the lower and middle canopies during

the early season and the understory during the mid season (data not shown), which may have contributed to the unclear pattern of vertebrate exclusion on caterpillars. However, we note that the four most abundant common insectivorous bird species observed in our study area includes one ground forager (Ovenbird), one species that gleans in the understory and lower crown (Black-capped Chickadee), and two species that glean in the main canopy (Red-eyed Vireo, Black-throated Green Warbler) (*Holmes, Bonney & Pacala, 1979*; *Robinson & Holmes, 1984*; *De Graaf, Tilghman & Anderson, 1985*).

Under most conditions, birds are able to limit forest arthropod abundance, especially that of herbivorous insects (e.g., *Holmes, Schutlz & Nothnagle, 1979*; *Marquis & Whelan, 1994*; *Philpott et al., 2004*). However, there has been no clear consensus when considering differences in bird predation pressure between vertical strata. *Van Bael, Brawn & Robinson (2003)* found the effects of bird predation to be higher in the canopy versus the understory, whereas *Boege & Marquis (2006)* found that bird exclusion reduced arthropod density by the same magnitude in both saplings and mature trees. Our study is the first to consider this question in temperate forests; for most arthropod groups in our survey, vertebrate predation pressure did not differ between vertical strata, including between understory saplings and three levels of mature canopy. However, we did find that vertebrate predation pressure was highest in the understory for spiders, the most abundant arthropods in our system. More distinct differences in vertebrate predation pressure between canopy strata might be seen in forest systems with higher densities of other arthropod groups.

The increased arthropod densities that we observed on caged branches may have been due to the exclusion of bats in addition to birds. Several tropical studies have found that using night cage treatments to exclude bats alone resulted in a greater increase in arthropod density than excluding birds only using day cage treatments (*Kalka, Smith & Kalko, 2008*; *Williams-Guillèn, Perfecto & Vandermeer, 2008*), suggesting that measurements of the effects of cage exclosures are actually measuring the effects of birds and bats combined. Bats in temperate forests seem to exhibit some degree of niche partitioning, such that some species are canopy specialists, some are sub-canopy and gap specialists, and some are habitat generalists (*Jung et al., 1999*). At least four species of bat have been recorded in the Morgan Arboretum (*Fabianek, Gagnon & Delorme, 2011*), and it is therefore possible that bat predation may have contributed to the mortality of arthropods on control branches.

We also cannot exclude the possibility that bat predation contributed to mortality in the overnight bait trials; however, we observed no birds feeding near baits in the daytime bait trails and therefore attribute all such predation events to invertebrates. These experiments indicated that invertebrate predators were most active in the understory and lower canopy. This pattern was primarily driven by ants, which accounted for the majority of predation events and whose abundance in our study decreased in abundance with increasing height from the forest floor. Ants may spend more time in the understory and lower canopy because of distance limits from colony nest sites on the ground (*Seifert, 2008*). However, studies in tropical forests have found ant density and predation rates to be higher in the canopy than in the understory (*Basset, Aberlenc & Delvare, 1992*; *Olson, 1992*), suggesting that ants will spend more time where their prey are more abundant. Ants are likely to have

been responsible for a majority of the overnight removal of baits as well; the dominant ant in our sites, *Camponotus pennsylvanicus*, is an aggressive and predominantly nocturnal forager, especially in late spring and early summer, though it forages in lower numbers during the day (*Klotz, 1984*).

This research supports a view of the forest canopy that is heterogeneous with respect to arthropod densities and refuge from natural enemies. Our study is the first to examine such vertical heterogeneity within temperate forests, and our findings support the view that vertebrate predators can have significant impacts on the densities of forest arthropods. It would be of interest to conduct future research on this topic in other forest types, locations, and years with higher densities of prey arthropods to better understand how these predator effects vary in space and time.

## ACKNOWLEDGEMENTS

We thank C Idziak and the Morgan Arboretum staff, JF Aublet, M Larrivée, and C Frost. SC Walker provided statistical advice and comments. We thank D Huber, M Ulyshen, and an anonymous reviewer for comments and suggestions on an earlier draft of this manuscript.

### Funding

This research was supported by the Natural Science and Engineering Research Council of Canada, the Canadian Foundation for Innovation New Opportunities Grant (Project no. 9548, to CM Buddle), and the Department of Natural Resource Sciences (McGill University). The funders had no role in study design, data collection and analysis, decision to publish, or preparation of the manuscript.

### Grant Disclosures

The following grant information was disclosed by the authors:
Canadian Foundation for Innovation New Opportunities Grant: Project no. 9548.
Natural Science and Engineering Research Council of Canada.
Department of Natural Resource Sciences (McGill University).

### Competing Interests

The authors declare no competing interests.

### Author Contributions

- Kathleen R. Aikens conceived and designed the experiments, performed the experiments, wrote the paper.
- Laura L. Timms analyzed the data, wrote the paper.
- Christopher M. Buddle conceived and designed the experiments, contributed reagents/materials/analysis tools.

**Peer**J

### Data Deposition

The following information was supplied regarding the deposition of related data:

1. Season total densities of arthropods (total and individual groups) on caged & control branches in four vertical strata.

http://dx.doi.org/10.6084/m9.figshare.749690

2. Results of a bait trial testing predation in four vertical strata.

http://dx.doi.org/10.6084/m9.figshare.749689.

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
