# Peer review of "Vertical heterogeneity in predation pressure in a temperate forest canopy"

_PeerJ, doi:10.7717/peerj.138_

## Round 0.1 · original submission · Minor Revisions

Thank you for submitting your MS to PeerJ.

First, a quick history of this MS. The MS was initially submitted and reviewed by two reviewers. The two assessments were "major revisions" and "reject." Because the reviews were mixed, I clicked the "reject" checkbox at that time and told the authors that they were most certainly welcome to resubmit the MS after revisions.

The authors did just that, and provided a lengthy and detailed rebuttal letter with their resubmission. The resubmitted MS (along with the rebuttal letter, etc.) went to one of the same reviewers as in the earlier submission and to a new reviewer as well. The result of the new review is again mixed, but on the positive side of the ledger. I.e., one "major revisions" and one "minor revisions." The reviewer who assigned "minor revisions" is one of the reviewers who also assessed the previous version of this MS. The reviewer who suggests "major revisions" provides a number of very good points, but all that I believe that the authors can address without much effort.

As such, I sort of wish that there were a "moderate revisions" option. But, since I do not believe that this needs to go out for re-review at this point, I am going to ask for minor revisions. The authors should *carefully consider* and revise and/or rebut as per the suggestions of the reviewer who asked for "major revisions." It should be noted that that reviewer mentions that the paper adds "new dimensions to the field," "achieve(s) impressive replication in space and time," and displays "impressive field identification of arthropod families." Thus this paper has been judged as contributing valuable information to the field and should be published following revisions/rebuttal.

Since three of the four reviews that this paper has undergone – which includes two of the three reviewers – have been designated as public reviews by the reviewers, I suggest that the authors consider making the reviews and other documents associated with this paper public as well. Although I do not know the PeerJ policy regarding this, I would also hope that the initially rejected MS, along with the reviews and rebuttal could also be published alongside the final version. This would allow other researchers to interact with substantial expert opinion on this paper and with the authors' thought process in the development of this study and subsequent report.

It would also be useful if the authors would publish the data (and any other useful outputs) associated with this study in a DOI-citable format with a service such as figshare, Dryad, or their own institution's repository (if one exists). The data should then be cited within the paper to link the data and the output for posterity.

Thanks again, and I look forward to the revisions.

·

Basic reporting

This paper reports an empirical manipulation of bird predators at four ‘stratifications’ or vertical layers of sugar maples in temperate forest. The authors measured impacts of the experiment on arthropod densities from six counts during one season. They detected an effect of birds overall, with higher densities inside cages, and particularly strong effects on several groups (eg spiders) and stronger effects in the understory than in the canopy. Although there are now numerous papers that have manipulated birds with cages or fencing to estimate impacts on arthropods, this study added several new dimensions: the aforementioned vertical element, a predation assay using sentinel insects, and a thorough survey of the birds encountered in the area. Because the authors could use a vertical lift to repeatedly access their units, they were able to achieve impressive replication in space and time.

The article is well written grammatically and in overall organization. It reads clearly with few typos, although there are some word discontinuities in fonts. Overall, the conceptual framing is adequate, but it also grasps for theoretical footing that is wholly beyond the study. Lines 30-32 and 191-192, this study does not test optimal foraging theory, which requires a lot more work than using sentinel insects. The conclusion that predators forage optimally should be dropped completely. And, the statement that predictions from optimality are consistent with Van Bael et al. does not make sense on multiple levels (32-34). First this sentence reads as if tropical forest canopies contain more insects, which is debateable and not addressed in that study. But Van Bael et al. also do not study optimality, so this idea should be removed for the same reasons as above. Finally, I question the logic of the prediction arising from these statements and the conclusions once the results are in. If one accepts that a greater bird effect was observed in the understory, than the understory is likely the area of greatest risk. If that is the case, why are there not higher arthropod abundances in the upper canopy where the effects are lower? Likely because predation risk is far from the only factor influencing their spatial distributions.

I’d recommend more care when using “predation risk”, as this is a per capita probability of getting eaten. The sentinel experimental speaks to risk whereas the community-level effect, as measured in the exclosure experiments, does not.

Experimental design

I was cheered to see an application of mixed models in the analysis. The tree identity as a random factor effectively controls for pseudo-replication that would be a problem if repeated samples were treated as fully independent. But this was not stated – I’d recommend language to that effect. However, I have the disquieting feeling that the results from these models were not really used, and that effect sizes (Cohen’s d) were the real metric used to derive conclusions. Given the explicit prediction that bird effects should depend on strata, I am puzzled by the statement that this statistical interaction was not that important after all (89-91), hence the Cohen’s d. A problem here is that these simple effect sizes did NOT control for repeated measures as far as I can tell. How were these actually calculated? Perhaps the results were first summed or averaged by tree identity, minimizing this issue, but this is not explicit. Moreover, even if the effect sizes are calculated appropriately, the rules of inference should be followed – confidence intervals that overlap each other cannot be considered different even if the means appear higher/lower. Thus the magnitude of the treatment effect was non-zero for all arthropods and spiders in the understory, but not for any other group (Figure 3), and the effect in the understory cannot be considered higher than at any other level (contrary to lines 150-155).

Model fitting and checking done according to Zuur … (83-84). This is a large multi-authored book. Need more specificity as to what this means. Similarly, with Crawley 2005 later on… single authored but still a vague description of what was done here.

Lines 104-105 on bird surveys, “no individual was recorded twice during the same day” – how would you know unless the birds are banded? And even if this was true, the same birds were clearly observed on different days, given the relatively small area surrounding the plots, and the large number of sightings of birds that ought to be territorial with limited movement (eg woodpeckers).

Why introduce and discuss the binomial generalized linear mixed models (128-130) if these are not mentioned at all in the results?
Replace “identification” with “identity” when discussing the random effect in mixed models (line 78, 129).

Validity of the findings

I was surprised by the prediction that arthropods should be more abundant and bird effects should be greater in the lower canopy. I would have predicted the opposite (as in van Bael) because the upper edge of the canopy intercepts the most light and is likely to be the most actively growing, and because insectivorous birds are typically active higher from the ground. Of course my hypothesis would not be supported here, but one should bear in mind, and be explicit in the discussion, that the authors used separate saplings in the understory, whereas the three upper layers used a single tree with three experimental sets. As Boege showed in several papers, ontogeny matters for the trees, for insects and for their predators, so this is a potential confounding variable that needs discussion. I appreciate that there was no other way because mature sugar maples won’t have much foliage in that zone, but that does not make the problem go away. One issue is for the mixed models, which labeled tree identity as a random effect, but the saplings and mature trees were not the same identities – how did you deal with that?

I would love to see the mixed models as more central to the results and the figures (and the discussion!). Table 2 quite clearly shows a consistent main effect for both cage exclusion and for stratum, but no interaction of the two. Figure 2 supports this, as the lines are mostly parallel among strata. You could also plot the coefficients from the models because these are model fits that takes the random effect into account, whereas I don’t know how the repeated measures were accounted for in the estimation of means and variance in this figure. But I don’t know what is meant in the sentence “selection procedures for the mixed effects models…..” (line 142-144).

I was surprised to see the muscoid flies in with the predators! Also, a pretty impressive field identification of arthropod families.

Additional comments

No additional comments

·

Basic reporting

see below

Experimental design

see below

Validity of the findings

see below

Additional comments

In my first review, my biggest concern about the paper was the possibility that the exclusion cages may have deformed the leaves on the enclosed branches. The authors now insist that this was not the case (lines 64-65) and have included pictures to show the installed cages in the field. Also, In their response to comments, the authors mention that “rigging” was used to hold the cages in place. I assume they mean that strings were used to support the weight of the mesh (i.e., to relieve the leaves) but this is still not described adequately in the methods. Because enclosing a branch within a mesh bag (made out of metal wire, no less!) without deforming the leaves is such a tricky proposition, I think a more detailed description of how this was accomplished is needed. I think these kinds of methodological details are important for the benefit of future researchers wanting to do something similar. Also, I feel that attention is still needed in lines 111-112 to clarify that the mealworms were actually punctured (i.e., it is possible to pin a larva against a trunk without puncturing it).

---

## Round 0.2 · accepted · Accept

Thank you for addressing the reviewers comments. Your responses and rebuttals to this round of review are fully satisfactory, and I recommend that this paper be accepted for publication in PeerJ.

Note that I have sent the PeerJ staff a message about the missing equations in the DOCX file of your rebuttal letter and have let them know that you supplied me with a PDF that provided those equations. If the reviewing/revision history of this MS is to published alongside the final article, I recommend that the rebuttal with the equations be included as a PDF.

Thanks for submitting your manuscript to PeerJ and for supporting the growth of open science.